# Simplified Modelling of Coupled Surface-Groundwater Transport Using a Subcatchment Mass Balance Approach

Alexander Hewgill Elliott [1,*], Channa Rajanayaka [2] and Jing Yang [2]

1   National Institute of Water and Atmospheric Research (NIWA), P.O. Box 11115, Hamilton 3214, New Zealand
2   National Institute of Water and Atmospheric Research (NIWA), P.O. Box 8602, Christchurch 8440, New Zealand; channa.rajanayaka@niwa.co.nz (C.R.); jing.yang@niwa.co.nz (J.Y.)
*   Correspondence: sandy.elliott@niwa.co.nz

**Abstract:** Catchment models based on steady-state mass balances enable rapid assessment of contaminant fluxes and concentrations in rivers. However, such models often focus on surface drainage, without taking groundwater into account. This paper presents a novel steady-state mass-balance catchment model that includes groundwater. The model incorporates a conceptual reservoir under each surface subcatchment, with lateral subsurface exchanges between adjacent reservoirs and vertical exchanges between the reservoirs and the surface drainage network. This leads to an easily solved coupled algebraic system of equations. The approach is demonstrated for nitrogen in a meso-scale catchment in New Zealand. Exchange coefficients were extracted from a full groundwater model, while recharge sources were obtained from separate hydrological and leaching models. Other parameters such as decay coefficients were determined through calibration. Although the exchange coefficients are generated from a detailed groundwater model, alternatives such as simple groundwater models or phreatic contours could be used instead. The effective decay parameters were different from what was expected, which is partly due to the model structure (for example, the assumption of complete mixing in each reservoir), but may also be due to input uncertainty. The applications demonstrated the successful deployment of a novel, simple, fast-running and flexible coupled surface-groundwater model.

**Keywords:** groundwater; surface water; water quality; model; coupling; nitrogen; model simplification

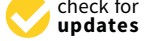

## 1. Introduction

Explicit representation of groundwater in models of catchment-scale contaminant generation and transport is desirable to enable more accurate representation of transport pathways and processes, especially in catchments with a regional groundwater system and for contaminants such as nitrate that are transported through groundwater. In simple catchment models, a surface-water-centric approach is often taken, without explicit representation of the groundwater system. As an example, the catchment model SPARROW (Spatial Regional Regression on Watershed attributes) [1,2] divides a catchment into subcatchments and an associated stream network, and contaminant sources are transported into the streams and down the stream network without taking groundwater into account explicitly. However, often a large part of the rainfall/irrigation and associated catchment load moves via groundwater. There may be losses or transformations of contaminants in the groundwater, and water recharged in one location may not enter the surface water in the local stream, but in some distant stream. One of the motivations behind the work reported in this paper is to add a groundwater component to surface-oriented catchment models.

Some coupled surface-groundwater models are available. For example, the Streamflow-Routing (SFR2) package [3] of the groundwater model MODFLOW (Modular Three-Dimensional Finite-Difference Groundwater Flow Model) [4] enables such coupling. Similarly, the model HydroGeoSphere [5] is an integrated surface-groundwater model. However, such models are complex to set up, time-consuming to run, and often neglect runoff

that reaches the stream by surface or near-surface runoff processes. The run time makes it difficult to undertake comprehensive uncertainty and scenario analysis. One approach to overcoming this run-time barrier is to develop meta-models [6], but usually that involves running the full model many times to develop an approximation of the system behaviour.

Another approach is to use simple steady-state or annual average flow and contaminant budgets, including a spatially distributed discretisation. Simple budget-based catchment models often do not take surface-groundwater interaction and groundwater routing into account (see [7,8] for recent reviews of catchment models). For example, the Water Framework Directive Explorer (WFD Explorer) [9] just applies contaminant attenuation factors between the source and the environmental receptor (such as the coast). Even dynamic (time-stepping) catchment contaminant models such as eWater Source [8] and SWAT (Soil Water Assessment Tool) [10] do not allow for movement and transformation of contaminants in regional groundwater systems.

To address these limitations, in this paper we present a spatially distributed budget-based catchment model including a simple representation of groundwater. A steady-state flow and contaminant budget approach is taken, which is simple and useful in practical applications that seek to quantify the source and transport pathways of contaminants. The model is spatially distributed, in that the catchment can be broken into many sub-catchments. Advantages of this approach include fast run times, compatibility with simple surface-oriented models and an intuitive conceptualisation.

This new model was applied in a case study of the Hauraki (3630 km$^2$) catchment in the North Island of New Zealand, which has a regional aquifer. Land use in the catchment is dominated by pastoral farming, leading to concerns regarding nutrient losses and their impacts on freshwater and estuarine receiving environments. Fully distributed groundwater models have been developed previously for the main aquifer in this system, which was useful for informing the simplified surface-groundwater model. Nitrogen was chosen as the contaminant of interest because it can travel through groundwater and is of ecological significance in the associated streams and estuaries.

This paper presents the concepts and mathematical development of this new model and demonstrates the setup and parameterisation of the model. It also demonstrates the development and application of a simple budget-based distributed meso-scale catchment model to represent coupled surface and groundwater transport for nitrogen, including consideration of the strengths and weaknesses of the approach.

## 2. Materials and Methods

### 2.1. Main Model Concepts and Assumptions

In this section, the main model concepts and assumptions are presented, while the mathematical formulation is presented in Section 2.2. The model spatial arrangement and mass exchanges are depicted in Figure 1. The catchment is represented as a set of subcatchments and associated stream segments and groundwater reservoirs. The segments and reservoirs are termed 'elements'. Only three subcatchments are shown in Figure 1, but in practice there may be thousands of subcatchments. Each subcatchment has a single associated stream segment, and stream segments are connected to make a drainage network, which is usually dendritic. Each subcatchment represents the land surface draining directly to the stream segment, which is determined by topographic elevations and gradients. A stream segment is a portion of a stream defined either between its beginning and a confluence, or between two confluences.

A reservoir is placed under each subcatchment, with the same boundaries (in plan view) as the subcatchment boundaries. Subsurface flow may occur between all its immediate neighbours. A reservoir may pass flow to or receive flow from multiple neighbours, and the subsurface flow directions can differ from the surface drainage directions. The potential for multiple subsurface directions means that flows cannot be simply accumulated from the top of the drainage network to the bottom; instead, mass transport equations are set up as a system of coupled equations.

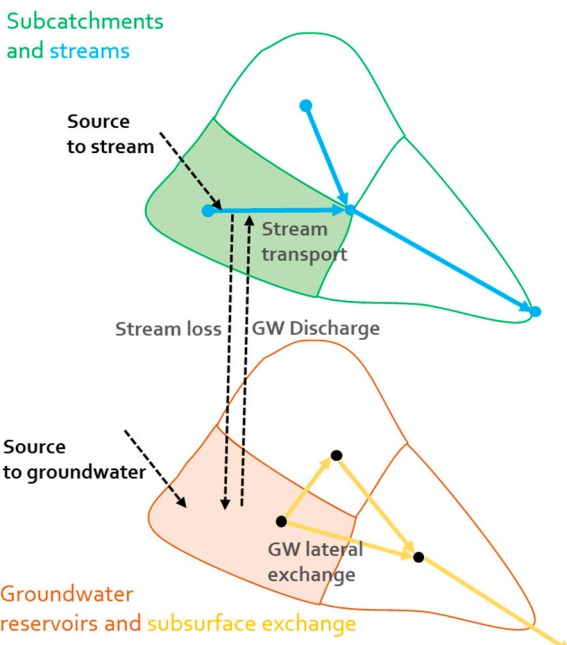

**Figure 1.** Schematic of the model spatial arrangement and exchanges. The shading indicates a matching subcatchment and its underlying reservoir.

Exchanges may also occur between a stream segment and its underlying reservoir. There can be losses from the stream to a reservoir (a 'losing stream') or discharge from the reservoir to the stream (a gaining stream).

A further key concept is that the outflows from an element are apportioned to its adjacent elements according to exchange coefficients, each of which represents the proportion of the total outflow occurring in the direction of interest. We used proportions rather than fixed flow rates because exchange coefficients would be more flexible. For example, if the sources were to be perturbed under a flow-change scenario, then new flow rates could be determined provided that the exchange coefficients remain the same. It may also be easier to estimate exchange proportions rather than absolute flow rates, for example, from phreatic gradients.

Each element can have sources, other than exchanges from other elements. Typically, we apportion the total water input (e.g., from rainfall and irrigation) from a subcatchment to (i) recharge to the reservoir and (ii) 'direct' losses to the stream (runoff).

The system of flow equations is developed by applying a flow continuity relationship for each element, whereby the total outflow from an element is the sum of flow sources and inflows from other streams or reservoirs. The equations are assembled into a matrix, which is solved using linear matrix methods (typically, sparse matrix methods).

A contaminant mass balance is developed for each element and assembled into a system of equations. It is assumed that each element is essentially 'well-mixed', in the sense that each outflow from an element has the same concentration as other outflows from that element. Hence, outflow contaminant loads are apportioned to adjacent elements in the same proportion as outflow flow rates, and the proportion of total mass outflow from an element that goes to an adjacent element can be determined using the relevant flow exchange coefficient. A proportion of the mass inflowing to an element can be lost from the system—typically from represented as 'decay'. Once the flows and loads are known, the flow-weighted mean concentration in an element can be obtained.

*2.2. Mathematical Formulation and Solution*

2.2.1. Flow Balance

The vector of total flows out of each element (volume per time) is denoted by $q$. This is represented as a stacked vector, where the top half is outflow from the streams and the

bottom half is the outflow from reservoirs. If element $q_k$ is the outflow from the stream in a subcatchment $k$, and there are *nseg* subcatchments (or segments) overall, then $q_{k+nseg}$ is the outflow from the reservoir below subcatchment $k$.

From mass balance, the total outflow from an element is the sum of inflows from other elements plus sources:

$$q = \text{A}^\text{T}q + b \tag{1}$$

where A is an exchange matrix and $b$ is a vector of sources. $A_{ij}$ is the proportion of the total outflow from element $i$ that is passed to element $j$, while $b_i$ is the source of inflow into element $i$ other than the inflow from other elements. $\text{A}^\text{T}$ is the transpose of A so that element $kl$ of $\text{A}^\text{T}$ is the proportion of outflow from element $l$ that goes into element $k$. The proportionality formulation ensures that the system equations are linear, is reasonable and ensures that the system can be solved if, for example, the sources change in a scenario.

The exchange matrix A can be partitioned into exchanges between: stream segments, F; lateral exchanges between adjacent reservoirs G; a diagonal matrix formed from the vector of stream loss proportions to reservoirs, $\beta$; and a diagonal matrix formed from the vector of reservoir discharges to streams, $\mu$:

$$\text{A} = \begin{bmatrix} \text{F} & \text{diag}(\beta) \\ \text{diag}(\mu) & \text{G} \end{bmatrix} \tag{2}$$

Rearranging Equation (1) gives the simple linear matrix equation

$$\text{B}q = b \tag{3}$$

where $B = \left(\text{I} - \text{A}^T\right)$ and I is an identity matrix. Equation (3) is a simple linear algebraic set of equations which can be solved readily, if values for the exchange proportions and other parameters are provided. Even though B can have large dimensions, it is sparse (there are sparse non-zero matrix elements) so that efficient sparse matrix solution methods can be used.

### 2.2.2. Contaminant Balance

The total flux out of an element (mass per time), $l$, is the sum of inputs multiplied by the decay within the element. It is assumed that all the outflows from an element have the same concentration, so that output flux from any element is in the same proportion as the outflows. Hence:

$$l = (\text{I} - \text{diag}(\delta))\,\text{A}^\text{T}l + (\text{I} - \text{diag}(\delta))d \tag{4}$$

where $d_i$ (element $i$ of the vector $d$) is the source flux entering element $i$ from its associated subcatchment (but excluding inflows from other elements), and $\delta_i$ is the decay fraction in element $i$, that is, the fraction of inflow to an element $i$ that is lost due to decay within that element. Rearranging Equation (4) gives

$$\text{D}l = (\text{I} - \text{diag}(\delta))d \tag{5}$$

where $D = \text{I} - (\text{I} - \text{diag}(\delta))\,\text{A}^\text{T}$. As with the flow balance, the mass balance can be solved with linear matrix methods.

The concentration flowing from an element $i$ can be obtained simply from the mass flux divided by the flow rate out of the element:

$$c = l/q \tag{6}$$

where the division is elementwise. This is a flow-weighted mean concentration, which may be different from the temporal mean or median concentration, and a factor may be used to convert between them [11].

The outflow from a stream into other streams, $q_s$ is the total outflow minus the losses to reservoirs:

$$q_s = (I - \text{diag}\begin{bmatrix} \beta \\ \mu \end{bmatrix})q \tag{7}$$

A similar equation is applied to obtain the load out of a stream to other streams, $l_s$:

$$l_s = (I - \text{diag}\begin{bmatrix} \beta \\ \mu \end{bmatrix})l \tag{8}$$

Element $ij$ of a matrix $D^{-1}(I - \text{diag}(\delta))$ is the proportion of the source to element $i$ that is transferred to element $j$. That is, $D^{-1}(I - \text{diag}(\delta))$ is a transfer matrix between sources and receptors. In general, the transfer matrix will not be sparse. This result is not needed for application of the model but is of interest in terms of relating the matrix formulation based on exchange coefficients to the intuitive concept of a transfer matrix.

This theory has been extended to allow for flow abstractions, two groundwater layers and for calculation of water age [12], but this paper has an abbreviated version for simplicity and clarity.

The sources can include point sources, direct runoff from the subcatchment into the local stream (the part of runoff that bypasses groundwater), and through land surface recharge from the subcatchment (excluding losses from streams). In the model applications, it was assumed that a proportion of $\alpha$ of the total water input on to the land surface of a subcatchment is recharged, the remainder entering the stream directly as runoff, and that the proportion can vary with subcatchment. If the concentration for direct runoff is the same as that in recharge, then the proportion of flow $\alpha$, can be applied to the total diffuse load source generated in a subcatchment to give the diffuse sources to the local stream and recharge.

While the model could use any user-specified decay fraction, we adopted the following approach. Decay within groundwater is assumed to be a first-order function of residence time $(T)$ in the reservoir, where the residence time is calculated from the outflow $(q)$, subcatchment plan area $(A_s)$ and an effective mixing depth $(H)$:

$$\delta = e^{-kT} = e^{-kA_sH/q} \tag{9}$$

where $k$ is a first-order time-based decay coefficient which may vary between reservoirs.

Decay within streams is assumed to be a first-order function of stream length, with the decay coefficient a power function of flow, as in the application of SPARROW to New Zealand [13]:

$$\delta = e^{-aq^bL} \tag{10}$$

where $a$ and $b$ are the coefficient and exponent respectively of the power function (determined by calibration), and $L$ is the length of the stream segment. The decay coefficient is allowed to vary with flow, and the exponent $b$ is usually negative, because streams with large flow rates generally have a smaller proportion of load lost to decay per unit stream length, compared to streams with small flow rates [13].

### 2.2.3. Solution Method

The matrices for flow and contaminant balance were set up by assembling sparse matrices from exchange coefficients, sources and decay parameters read in from a text file. Text files were used for input and output to make for easier interfacing with general parameter calibration software PEST++ (Parameter ESTimation ++) software [14,15] which uses text files. The model was implemented in the Python language (ver 3.5), with use of the sparse linear algebra solver called spsolve within the open-source SciPy library (ver 1.7) [16] which in turn uses the high-performance numerical library SuperLU [17]. Model code and example input and output data are included in a set of files in the Supplementary Materials.

*2.3. Exchange Coefficients*

The model requires exchange coefficients between groundwater reservoirs, and between streams and groundwater, and between streams. The exchange between stream segments is defined by the network topology (all the outflow apart from stream losses goes to the next downstream segment).

In the simplest case (apart from the case where there is no recharge), all the groundwater recharge in a subcatchment emerges into the stream in the same catchment, and there is no stream loss. All the subsurface exchange coefficients (elements of G) are then zero, and the stream loss coefficients $\beta$ are zero. This approach was used in our case study in areas where there is no regional groundwater (e.g., hilly subcatchments). Exchanges between adjacent reservoirs (elements of G) were simulated for the area covered by the regional groundwater model that encompasses a large proportion of the case study area.

Another simple approach is for the direction of subsurface exchanges to mirror the direction of the overlying surface drainage network. So, if a stream in subcatchment A flows into the stream in subcatchment B, the subsurface reservoirs also drain in the same direction (from the reservoir under A to the reservoir under B). This approach still requires estimation of exchanges between streams and groundwater, which could be informed by stream gauging measurements.

The most general case would be to allow exchange coefficients to be varied independently and estimated using high-dimension parameter estimation techniques [14,15]. We found that approach introduced complexity associated with ensuring continuity (sum of exchange coefficients sums to 1) and required a massive number of runs in a trial application, so that approach was not ultimately used in the case studies.

Derivation of Exchange Coefficients from MODFLOW Groundwater Model Results

The approach used in the case studies, for areas within the regional groundwater zone, was to derive exchange coefficients from results of a full groundwater model. In the case study a groundwater model based on MODFLOW had already been developed and are documented [18]. Derivation of the exchange coefficients required considerable post-processing of the groundwater model results.

Losses from the streams to reservoirs were determined directly from losses per stream length in the MODFLOW output. The MODFLOW stream network was based on the stream reaches of the catchment model, but only on Strahler order 3 and larger of the stream network in the catchment model. Since it is not easy to downscale losses from large streams (i.e., higher Strahler order) to smaller streams (i.e., lower Strahler order), losses were limited to the larger streams.

Flow across the boundary between adjacent reservoirs was determined based on spatial downscaling of vertically averaged MODFLOW outputs. First, the three-dimensional (3-D) MODFLOW fluxes were converted to a 2-D vertically averaged flux by summing fluxes over the MODFLOW vertical layers. The groundwater model used 1 km by 1 km grid cells, which are larger than the scale of subcatchments and do not align with the irregular subcatchment boundaries (see Figure 2, which is taken from part of the Hauraki catchment). To address this mismatch, the groundwater model plan grid was subdivided into finer resolution cells (10 m by 10 m grid cells, although the fine cells were ten times smaller than that depicted in Figure 2). Each of the fine cells was allocated to a subcatchment based on the location of the centre of the fine cell. The flow across each of the groundwater subcatchment boundaries was then determined, using linear interpolation of velocities from the original MODFLOW grid. The total flux across the boundary was then determined, and the fluxes across all boundaries were then accumulated for each element to determine the outflows as a proportion of the inflows.

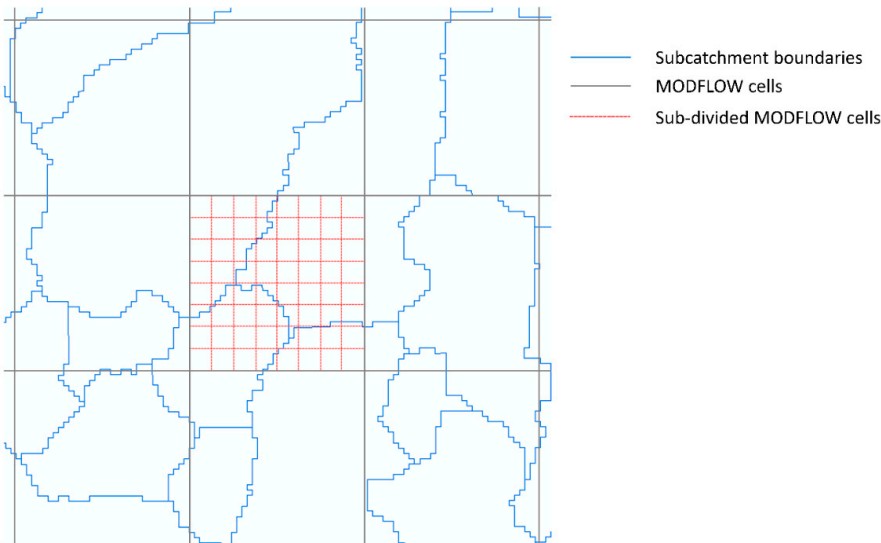

**Figure 2.** An illustration of the overlapping of subcatchment boundaries, original MODFLOW cells and fine-resolution subdivisions of the MODFLOW cells, taken from part of the Hauraki case study catchment.

*2.4. Hauraki Case Study*

The Hauraki case study addressed the catchment of the south end of the Haruaki Gulf in New Zealand (Figure 3) which is dominated by the catchment of the two main rivers, the Piako River and the Waihou River. The geology is predominantly ignimbrite [19], with well-drained soils in the upper catchment. Nitrogen loads entering the Gulf are of concern due to potential marine eutrophication [20], while concentrations in the streams are of concern due to excessive growth of macrophytes and epiphytic periphyton. The catchment has been studied as part of a research program on surface-groundwater model simplification and uncertainty [12]. The total catchment area is 3630 km$^2$, which was broken into 7925 subcatchments and their associated stream segments.

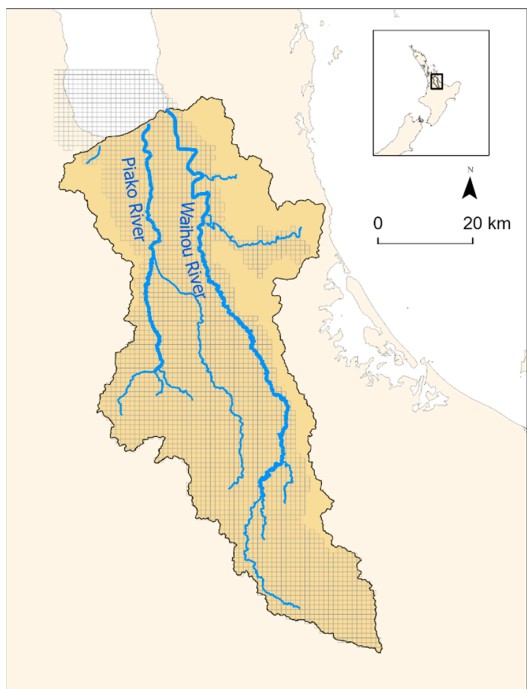

**Figure 3.** Location of the Hauraki case study. Grey grid lines show MODFLOW groundwater model cells. The MODFLOW domain only covers a part of the case study area. The blue lines show the main rivers and tributaries.

Key data sources used in the model are summarised in Table 1. Subsurface exchange coefficients and surface-subsurface exchanges were determined as described in Section 2.3. Subsurface first-order decay coefficients used a map of shallow groundwater oxic status class (Figure 4) which was developed with a spatial regression model [21] and provided three classes of oxic status. A separate decay coefficient was used for each decay class (through calibration) and the representative decay parameter for a subcatchment was the area-weighted decay parameter based on the areas of the class within the subcatchment. In areas where the decay class is not available, the value for mixed oxic status was used.

**Table 1.** Key data sources used for the Hauraki case study.

| Item | Source |
|---|---|
| Surface subcatchments and streams | River Ecosystem Classification (REC) drainage network version 2.4 [22] |
| Direct runoff, recharge and runoff apportionment | Hydrology model TopNet [23,24] |
| Diffuse nitrogen sources | Overseer catchment model [25,26], as applied by DairyNZ [27] |
| Point sources | WRC monitoring summary [28] |
| Groundwater model (used for exchange coefficients) | Steady-state model by GNS Science MODFLOW-NWT-SFR [18] |
| Redox zones | Shallow groundwater redox status model [21] |
| Calibration loads | Derived from monitoring data by WRC [28] |

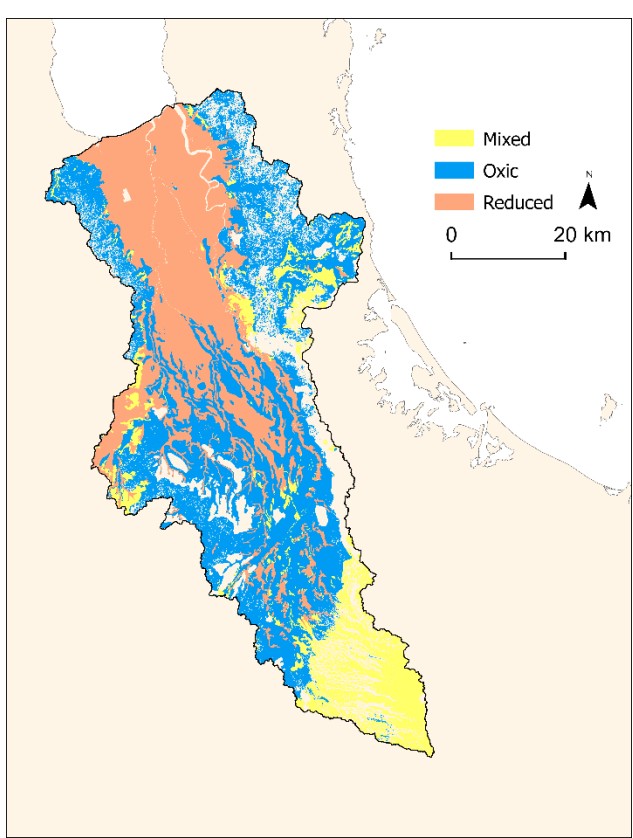

**Figure 4.** Map of groundwater oxic classes for the Hauraki [21].

Key remaining parameters include (i) the first-order subsurface decay rate coefficient for each of three groundwater oxic state classes, (ii) a stream decay coefficient and (iii) an effective mixing depth for each of a number of zones associated with monitoring sites.

Monte Carlo methods were used to calibrate model parameters, using the SWEEP facilities of the PEST++ (Parameter ESTimation ++) software [14,15]. The methods sample parameters from a prior distribution, and the best-performing parameters sets are selected to arrive at a 'conditioned' set of model parameters (which we have termed 'posterior' parameters, even though the model estimation does not use Bayesian methods for which the term posterior originates). Methods used for establishing the prior distribution, and the shape and parameters for the distributions, are summarised in Table 2. About 10,000 parameter realisations were used (the model was run 10,000 times). Stream nitrogen loads at 12 sites derived from monthly water quality monitoring measurements and continuous flow records [28] were used for assessing the goodness of fit. The sum of absolute deviations in load was used as the fit metric. The 'posterior' parameter distribution was taken from the best 200 runs (smallest values of the fit metric), rather than defining an acceptable value for the metric.

**Table 2.** Key parameters other than exchange coefficients, methods of determining their prior distribution and the prior distribution, for the Hauraki case study.

| Parameter | Method of Determining Prior Values | Parameter Baseline Value and Distribution |
|---|---|---|
| Subsurface decay coefficients | Global oxic, anoxic and mixed nitrogen first order removal rate coefficients. Baseline values from independent advice (Murray Close, ESR). Distribution was biased less than initial estimates after initial model exploration. | Oxic = 0.0000365 yr$^{-1}$ (0.0000001–0.00001) Mixed = 0.9993 yr$^{-1}$ (0.001–0.9999) Anoxic = 1 yr$^{-1}$ (0.01–1) Distribution: Uniform. |
| Mixing depth (water equivalent depth) | Initial depths were based on the thickness of the saturated layer of the MODFLOW model. However, that approach gives large mixing depths and resulted in high decay rates. Hence, the mixing depth was set as no more than 2.5 m. | ≤2.5 m (0.5–2.5) Distribution: Uniform. Values adjusted for zones associated with each of 18 river monitoring stations. |
| Stream decay coefficient | Based on CLUES [13]. Exponent set at −0.7. | 0.01 km$^{-1}$m$^{2.1}$s$^{-0.7}$ (0.0001–0.02) Distribution: Uniform |
| Concentration ratio for surface runoff (ratio of direct discharge to groundwater recharge concentration) | Assumed constant | Set to 1, not varied. |

## 3. Results

*Hauraki Case Study*

The model ran rapidly (3 s for a single iteration on a personal computer, including input-output), the numerical solution was stable and the system conserved mass. This result was not unsurprising given the simple linear formulation. Due to the rapid run-time, it was feasible to run 10,000 Monte Carlo iterations on a four-core workstation in 4.5 h. The most complicated part of the setup was extracting exchange coefficients from the groundwater model.

A numerical difficulty occurred when there were cycles in the flows, when the exchange coefficients were such that all the outflow from an element returns to that element. In this situation there is not a unique solution to the flows and the matrix solver fails. This unusual situation, which is actually a poorly specified model, only arose when merging the groundwater model domain with the full model domain; additional forensic code was required to detect and correct the source of the cycles, based on examining increments of flow with an iterative matrix solver.

Example results comparing measured and predicted loads before and after parameter conditioning are shown for the downstream sites in Figure 5. Parameter conditioning generally shifts the distribution of predictions closer to the values estimated from measurements ('measured' values) and narrows the distribution of parameters. This reduction in bias is associated with selecting the best-performing set of parameters realisations. Despite the general improvement, there was some deterioration for some of the smaller catchments such as the Waitakaruru River catchment, because the parameters such as the global decay coefficients were dominated by sites with larger loads. While the model performance for small sites was sometimes poor, this does not have much influence on the overall result due to the choice of fit metric. Alternative fit metrics could have been used to place more emphasis on smaller sites (for example, by normalising the loads to measured loads in the fit metric). Other sites (not shown) had a negative bias, which was retained after parameter conditioning.

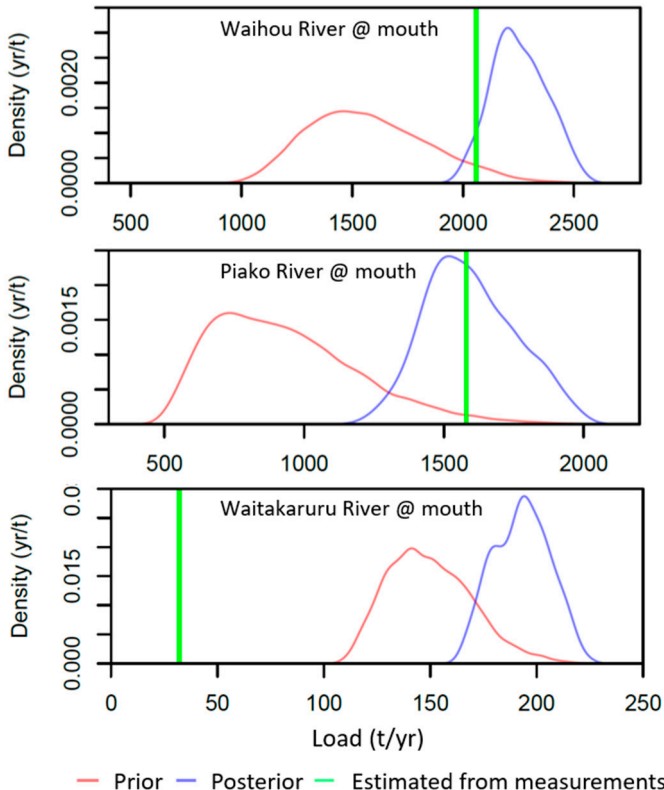

**Figure 5.** Example comparisons between measured and predicted loads before and after parameter conditioning. The curves are the smoothed probability density distributions of the predictions.

The distribution of groundwater decay parameters after conditioning (Figure 6) shows that the decay coefficients were considerably less than initially estimated from independent advice. The decay coefficient in oxic groundwater was very small, but despite the small decay, the predicted loads were less than measured in some of the subcatchments in oxic zones, suggesting that the leaching sources provided from a separate model may be too high. There is no mechanism in the model for creating additional source loads. These small decay coefficients are also influenced by the large residence times associated with the assumption of complete mixing in the conceptual reservoirs. For example, with a shallow effective depth (equivalent water depth) of 2.5 m, the residence time for a typical recharge of 0.1 m/y would be 25 years, leading to attenuation by a factor of 0.082 (91.8% reduction) for a decay coefficient of 0.1 $y^{-1}$. We also note that the independent estimates are actually derived from calibration of different models, rather than from direct measurement of process rates, so the independent estimates are not themselves definitive. No measurements of decay rates have been made in this catchment, though.

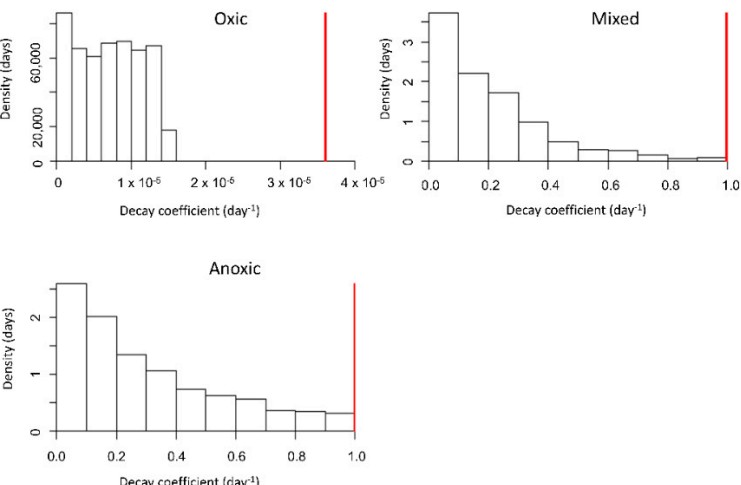

**Figure 6.** Distribution of groundwater decay coefficients for oxic, mixed and anoxic conditions. The posterior distribution is the histogram bars, while the prior value is the red line.

As an example, application of the approach, a load reduction scenario was run whereby all sources in the catchment were reduced by 20%. The resulting concentration is shown in Figure 7. The distribution of concentrations is biased low compared with the measured median, which is in part due to under-prediction of the load at that location. Additional errors may arise due to estimating the median concentration from the mean annual load. The load reduction leads to a 20% reduction in concentrations, due to the linear model formulation and the assumed uniform reduction of sources. Despite the reduction and uncertainty in predictions, the concentrations remain above the target level of 0.5 mg/L (an interim target, which is likely to be replaced). The bias in concentrations suggests that bias correction may be needed for comparing model predictions with absolute concentration targets.

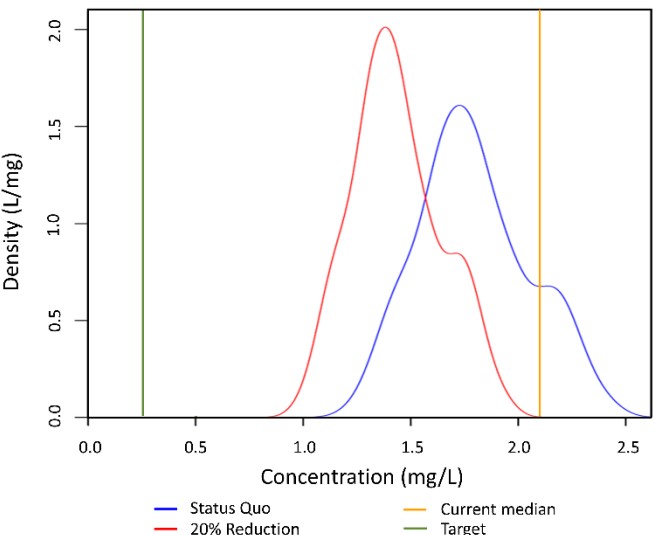

**Figure 7.** Example results for the probability density distribution of total nitrogen concentrations for a hypothetical load reduction scenario for Piako River at Paeroa-Tahuna Road Bridge.

## 4. Discussion

In the application of the model, exchange coefficients were determined by extracting information from a 3-D groundwater model. There are some limitations with that approach, such as a mismatch of scales and spatial discretisation. For example, the groundwater model only included stream losses for Strahler order 3 streams, whereas it would be desirable to have losses for smaller streams in the simplified subcatchment-based model. Converting from the coarse rectilinear groundwater model grid to the irregular subcatch-

ment boundaries required considerable manipulation of the groundwater model results. The groundwater model could have vertical variations in flows—for example, fast shallow components—which had to be averaged vertically for simplified subcatchment-based model, which then led to long residence times and associated high decay. This can be countered to some degree by restricting the depth of the model to an effective depth that is calibrated, losing some aspect of the physical interpretation of the formulation and leaning more towards a conceptual formulation.

Lumped-parameter dynamic modelling of a subcatchment within the Piako catchment by Woodward and Stenger [29] demonstrated that long residence-time pathways (such as deep groundwater) were associated with low concentrations, whereas shallow fast pathways had high concentrations, suggesting the desirability of separating shallow and deep groundwater pathways. The model formulation presented in this paper has been extended to include a deep groundwater layer [12], but was not explored in the model application due to the need for additional parameterisation.

The model formulation allows for extracting exchange coefficients from a variety of sources. There are alternatives to using a fully distributed groundwater model for evaluating the exchange coefficients. As mentioned in Section 2.3, in some cases it could be assumed that all the recharged groundwater exits into the local stream, or that subsurface flow mirrors surface pathways. A further approach would be to extract the exchange coefficients from a simplified groundwater model developed recently for New Zealand [30], which could be explored in future work. In some cases, piezometric surfaces may be available, and they could be used for estimating subsurface exchange coefficients, although it would still be difficult to evaluate stream losses and groundwater discharge.

An alternative approach to model simplification in the Hauraki catchment by White et al. [31] used a linear model emulator of a fully distributed steady surface-groundwater model coupled with a transient nitrate transport model. The emulator reasonably assumed linearity between source loading and stream concentrations. The sensitivity of concentrations to sources (a response matrix) was determined by undertaking a full model run for each grid cell, changing the source in that cell and noting the response at each location of interest. In the model application, 3160 grid cells each of 1 km$^2$ area were used, requiring 3160 model runs to represent the sensitivity of concentrations to source loadings. The linear formulation was convenient for coupling with an economic optimisation model for source load reduction including parameter uncertainty. A similar linear approach could be taken using the simplified subcatchment-based model presented in the current paper, building on the transfer matrix concept discussed in Section 2.3.

The model formulation and application only addressed some aspects of uncertainty. For example, it was assumed that the nutrient sources and the split between recharge and runoff direct to the local stream are known deterministically; uncertainty in the measured load was not accounted for; and the structural uncertainty is not addressed (for example, the assumption of mixing in each subsurface reservoir).

Calibration was only done for load estimates in the streams. Some groundwater concentration estimates are available, and the model can predict a concentration in groundwater, but measured concentrations are likely to vary with depth and at fine spatial scales not represented in the model, so that comparisons between modelled and measurements are not likely to be of much value.

The model has been extended to predict the water age [12]. That extension was not applied in the case study, but may offer opportunities in the future for constraining parameters such as the effective mixing depth better, if measurements of water age are available.

The model formulation introduces the potential for parameter interaction. For example, decay coefficients are likely to interact with the effective groundwater depth and source parameters. Such interactions make parameter calibration difficult (for example, requiring more parameters realisations to explore the model behaviour).

The model is steady state, focusing on long-term average water and mass balances. It is possible to use flow-concentration relationships in conjunction with a measured or

simulated time series to disaggregate concentrations over time (as done recently in the Hauraki catchment [32]), but that introduces uncertainties associated with the rating curve form and evaluating the rating curve for un-measured locations. A potential avenue for future work would be to extend the spatial constructs of the model presented in this paper (a conceptual reservoir under each subcatchment, with exchange proportions between adjacent reservoirs) but taking account of dynamics (for example, thorough storage-discharge relationship for each reservoir), possibly building on conceptual dynamic groundwater extensions from other hydrological models (e.g., [23]).

## 5. Conclusions

The novel simple model presented in this paper provides a simple and flexible approach to adding regional groundwater to steady-state models for flow and contaminant transport models that use spatial constructs of subcatchments and an associated drainage network, which are commonly used for representing surface water systems.

The calculations proved to be straightforward to set up and solve, with rapid run-times (<10 s) for a meso-scale catchment with 7925 subcatchments. Such rapid run-times allow for parameter calibration methods that involve many model realisations. The simple formulation would be easy to incorporate in couple catchment-economic models.

The model entails the specification of many exchange coefficients, which define the proportion of the total outflow that passes to adjacent elements (reservoirs or stream segments). In the example application, the coefficients were derived from an existing 3-D groundwater flow model, but alternatives are available, such as basing flow directions on groundwater contours.

The trial application showed model biases even after calibration. Part of the biases could be due to relying on separate models for some of the inputs, such as nutrient sources, recharge fractions and locations of reducing groundwater conditions, which could not be compensated for by adjusting other parameters. The model also incorporates some conceptual parameters, such as the effective mixing depth, which are difficult to determine with limited observations and when there is inherent co-variation with other parameters such as decay.

Overall, the approach presented in this paper provides some new concepts that could be further developed in future models, and are of some immediate use for rapid assessment of contaminant fluxes in coupled surface-groundwater systems, provided that challenges with parameterisation are catered for.

**Supplementary Materials:** The model code and example dataset are available online at https://www.mdpi.com/article/10.3390/w14030350/s1.

**Author Contributions:** Conceptualisation, software and manuscript lead: A.H.E. Hauraki case study setup, calibration and analysis and manuscript contribution: C.R. Groundwater model analysis and manuscript contribution: J.Y. All authors have read and agreed to the published version of the manuscript.

**Funding:** This research was funded by the Ministry of Business and Innovation and Employment Smart Models for Aquifer Management Programme Contract No. C05X1508.

**Institutional Review Board Statement:** Not applicable.

**Informed Consent Statement:** Not applicable.

**Data Availability Statement:** Not applicable.

**Acknowledgments:** This research was funded by the Ministry of Business and Innovation and Employment Smart Models for Aquifer Management Programme. We thank DairyNZ for providing leaching data for the Hauraki catchment, Waikato Regional Council for providing water quality and flow data, Jeremy White from GNS Science for providing MODFLOW model results for the Hauraki catchment, and Murray Close from ESR for providing oxic status maps and initial parameter estimates. We thank Christian Zammit from NIWA for providing recharge estimates.

**Conflicts of Interest:** The authors declare no conflict of interest.

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
