# Peer review of "Simplified Modelling of Coupled Surface-Groundwater Transport Using a Subcatchment Mass Balance Approach"

_water, doi:10.3390/w14030350_

Round 1

Reviewer 1 Report

This study combined hydrogeology to improve the model for evaluating contaminant fluxes and concentrations in rivers, and verified it with an example of nitrogen pollution in a river basin in New Zealand. The process of model improvement is complete, the discussion is comprehensive, and the effect of the improvement is truly and meticulously explained, but some parts of the writing skills should be improved. Therefore, I think the manuscript can be accepted after simple revision. The following are detailed opinions:

The title can simply mention the verification area as an example

The quality of the abstract can be further improved. I think it is possible to briefly describe the prediction results of the example to highlight the effect and progress of the improved method

Please name Figure.2 with concise content. Its original name is more like an explanation. Please put it in the text or express it in the form of a legend.

In Results, the author uses three rivers to illustrate the prediction effect after adjusting the parameters. How to ensure the representativeness of the three rivers? How does the deterioration of smaller catchments affect the overall result? Please discuss.

Line 353-362, the reduction of decay coefficients should not only be compared with initially estimated from independent advice. If conditions permit, please join the experimental comparison of traditional methods to reflect the superiority of the improvement.

Author Response

This study combined hydrogeology to improve the model for evaluating contaminant fluxes and concentrations in rivers, and verified it with an example of nitrogen pollution in a river basin in New Zealand. The process of model improvement is complete, the discussion is comprehensive, and the effect of the improvement is truly and meticulously explained, but some parts of the writing skills should be improved. Therefore, I think the manuscript can be accepted after simple revision. The following are detailed opinions:

The title can simply mention the verification area as an example.

- We have considered the reviewers suggestion to add the verification area in the title as suggested by the reviewer, but have decide that it would be better not to add that information because the journal prefers concise titles. The title is already 12 words long and adding more words such as ‘with an application in the Hauraki catchment, New Zealand’ would make the title long. The demonstration/verification site is mentioned in the abstract, so will come to the attention of the reader pretty quickly without being in the title.

The quality of the abstract can be further improved. I think it is possible to briefly describe the prediction results of the example to highlight the effect and progress of the improved method

- We would like to add the information. However, the abstract is limited to 200 words, and is currently 199 words long. Therefore there is not space to add specific quantitative results for the example application. We have instead preferred to describe the new model briefly, including how it is parameterised, with only broad findings from the example application.

Please name Figure.2 with concise content. Its original name is more like an explanation. Please put it in the text or express it in the form of a legend.

- OK, done.

In Results, the author uses three rivers to illustrate the prediction effect after adjusting the parameters. How to ensure the representativeness of the three rivers? How does the deterioration of smaller catchments affect the overall result? Please discuss.

- OK, some discussion has been added.

Line 353-362, the reduction of decay coefficients should not only be compared with initially estimated from independent advice. If conditions permit, please join the experimental comparison of traditional methods to reflect the superiority of the improvement.

- Some comments on this have been added to the paper, pointing out that the values from independent advice are not definitive, and ideally that direct measurements would be made, but that we do not have such measurements in this catchment.

Reviewer 2 Report

Line 127-128  – please revised the phrase

 In equation 4 the second „diag„ should be italic

Line 168-169 – are presented  „??„ and „??„ but there are missing from the equation (4).

Line 172 – please correct „D=I- (I−????(?)) AT„

Line 176 - please correct the word ”weighed”

Line 208 - what exactly characterizes parameters a and b??? At what they do refers??

Line 216 - correct the word ”spsolve” please.

Author Response

Line 127-128  – please revised the phrase. 
- OK, revised.
In equation 4 the second „diag„ should be italic
- Have revised diag so that is it consistently non-italicised throughout
Line 168-169 – are presented  „??„ and „??„ but there are missing from the equation (4).
- d and δ were in the equation. Have clarified text to make it clear that d¬ is the i’th element of d
Line 172 – please correct „D=I- (I−????(?)) AT„
- OK, formatting fixed
Line 176 - please correct the word ”weighed”
- OK, done
Line 208 - what exactly characterizes parameters a and b??? At what they do refers??
- a brief explanation has been added.
Line 216 - correct the word ”spsolve” please.
- spsolve is the name of a function. The text has been modified to make this clearer.